


# Limitations of Demand- and Pressure-Driven Modeling for Large Deficient Networks

Mathias Braun[1], Olivier Piller[1], Jochen Deuerlein[2], and Iraj Mortazavi[3]

[1]IRSTEA, 50 Avenue de Verdun, 33612 Cestas, France
[2]3S Consult GmbH, Albtalstrasse, 76137 Karlsruhe, Germany
[3]CNAM, EA-7340-M2N-Modelisation Mathematique et Numerique, 75003 Paris , France

*Correspondence to:* Mathias Braun (mathias.braun@irstea.fr)

**Abstract.** The calculation of hydraulic state variables for a network is an important task in managing the distribution of potable water. Over the years the mathematical modeling process has been improved by numerous researchers for the utilization in new computer applications and the more realistic modeling of water distribution networks. But, in spite of these continuous advances, there are still a number of physical phenomena that cannot be tackled correctly by current models. This paper will take a closer look on the two modeling paradigms given by demand and pressure driven modeling. The basic equations are introduced and parallels are drawn to the optimization formulations from electrical engineering. These formulations guarantee existence and uniqueness of the solution. One of the central questions in the French and German research project ResiWater is the investigation of the network resilience in case of extreme events or disasters. Under such extraordinary conditions where models are pushed beyond their limits we talk about deficient network models. Examples of deficient networks are given by highly regulated flow, leakage or pipe bursts and cases where pressure falls below the vapor-pressure of water. These examples will be presented and analyzed on the solvability and physical correctness of the solution with respect to demand and pressure driven models.

## 1 Introduction

Calculating the flow in hydraulic networks has a long history starting with the work presented by Cross (1936). Today more than ever it is an important component in managing the distribution of potable water. Originally being developed for planning and sizing of water distribution networks (WDNs), the applications have since been extended to areas like sensor placement, leakage reduction, water security and online system management (SMaRT-Online[WDN]). In the application for systems with inadequate capacity or pipe failure, the classical demand-driven modeling (DDM) approach is stretched to its limits. This paper is published as part of the French-German project ResiWater. The objective of the ResiWater project is the evaluation and improvement of resilience and reliability of water distribution networks in the presence of extreme events. These events may be triggered by considerable technical accidents or natural disasters. The objective of this paper is to take a closer look on advances done in the area of hydraulic network modeling with respect to the robust, pressure-driven modeling for cases of lost connectivity in parts of the network. These zero-flow conditions are especially demanding, since the non-linear flow/tension problem is ill conditioned. First, in section 2 the classical demand-driven modeling approach will be presented starting from the basic





methods given by Cross (1936). In his approach the outflows at demand nodes are given as fixed boundary conditions. Then the major developments in variational methods and optimization by authors like Birkhoff (1963), Collins (1978) and Carpentier et al. (1985) are presented leading to the complete definition of primal dual problems. Second, in section 3 the development of an alternative modeling approach also known as pressure-driven modeling (PDM) is presented. Based on the fact first voiced by Wagner et al. (1988) that outflow at demand nodes is not a fixed, but rather a pressure dependent boundary condition, a number of approaches have been developed. Several authors introduced functions called Pressure-Outflow relationship (POR) to determine the actual flow based on the available pressure. Early approaches as presented by Bhave (1981) use a POR in an iterative approach to solve a series of DDM problems while adjusting the demands to be compatible with the pressure. Piller and Van Zyl (2007) presented a mathematical formulation of the pressure-driven model that does not rely on the definition of any Pressure-Outflow relationship. Instead, the authors use modified mass-balance constraints at consumption nodes to allow reduced demands in case the pressure is insufficient Piller and Van Zyl (2009). Finally, in section 4 some conclusions are given on the limitations of existing methods and an outlook is presented on possible improvements that are studied in the project ResiWater. In particular, the project deals with challenges that result from situations where the connectivity of the network is lost due to massive system failures caused by extreme events that often lead to insufficient pressure conditions even in the remaining system.

## 2 Demand Driven Modeling

### 2.1 Basic Hydraulic Equations

In hydraulic modeling the simplified topological structure of a water distribution network is described by a directed graph. This graph represents pipe sections as a link and pipe junctions as nodes. The mathematical description of this graph is given by the incidence matrix $\mathbf{A}$ and is defined as:

$$A_{i,j} = \begin{cases} -1 & \text{, if pipe } j \text{ enters node } i \\ 0 & \text{, if pipe } j \text{ is not connected to node } i \\ +1 & \text{, if pipe } j \text{ exits node } i. \end{cases}$$

Water distribution networks have a looped structure and the system state is described by the potential at the nodes (head) and the current on the links (flow). The system equations are given by the following sets of equations. First the mass balance at every node

$$\mathbf{A}\mathbf{q} + \mathbf{d} = \mathbf{0} \tag{1}$$

and second the energy equation.

$$\Delta\mathbf{h}(\mathbf{r},\mathbf{q}) - \mathbf{A}^T\mathbf{h} - \mathbf{A}_f^T\mathbf{h}_f = \mathbf{0} \tag{2}$$



Here the incidence matrix is divided into two parts. $\mathbf{A}$ describes the part of the network that only contains simple nodes, whereas $\mathbf{A}_f$ describes the nodes with fixed potential like reservoirs or tanks. The vectors for flow $\mathbf{q}$ and head $\mathbf{h}$ for the unrestrained links and nodes are the variables describing the system state. Boundary conditions can be defined by means of two parameter vectors. First, the nodal discharges at simple nodes, also termed the demand vector $\mathbf{d}$ and second, the heads at

nodes with a fixed potentials $\mathbf{h}_f$. Finally, $\Delta\mathbf{h}(\mathbf{r}, \mathbf{q})$ in the energy equation describes the head-loss along a pipe due to friction. Head-loss in general is a non-linear function of a friction coefficient $\mathbf{r}$ and flow $\mathbf{q}$. A generalized form of the relation between head-loss and flow is defined by Nielsen (1989) as follows:

$$\Delta\mathbf{h} = \mathbf{E}\mathbf{q} \qquad \text{with:} \qquad \mathbf{E} = \mathbf{r}\,|\mathbf{q}|^{\alpha-1}. \tag{3}$$

The inverse relation is defined as:

$$\mathbf{q} = \mathbf{E}^{-1}\Delta\mathbf{h} \qquad \text{with:} \qquad \mathbf{E}^{-1} = \mathbf{r}^{-\frac{1}{\alpha}}\,|\Delta\mathbf{h}|^{\frac{1-\alpha}{\alpha}}. \tag{4}$$

The constant $\mathbf{r}$ contains information based on the pipe parameters like diameter, length, roughness and more. Common choices for its calculation, depending of the country, are the Prandtl-Colebrook, Darcy-Weisbach or Hazen-Williams formula.

## 2.2 Content and Co-Content Model

Based on the works of Cherry (1951) and Millar (1951) in 1951 for electrical systems, Collins (1978) introduced a primal and

dual formulation of the hydraulic problem based in mathematical optimization theory. These formulations are also known as the Content and the Co-Content Model. However, minimization of the Co-Content Model is equivalent to a variational approach introduced by Birkhoff (1963).

For linear systems Maxwell's theorem states, that the distribution of current or flow which gives a minimum value to the function

$$F(q, h) = W - 2P_h \tag{5}$$

is the only one consistent with Kirchhoff's equations. Here $W$ defines the total energy lost in the system and $P_H$ the total power taken from the fixed potential nodes. Millar (1951) extends the theoretical framework for nonlinear systems by dividing the energy-loss into two components called the content $G$ and co-content $J$. The sum of content and co-content is equal to the total energy lost in the system.

$$W = G + J \tag{6}$$

For linear systems $G$ and $J$ are equal, thus allowing to calculate the system state by equation (5). In nonlinear systems the content and co-content are defined by the integral over an elements characteristic. This characteristic is a function of head-loss and flow and for the scope of this paper is given by $\Delta h - eq = 0$. Using these definitions a primal and dual optimization problem can be formulated for the hydraulic model with the primal functional $C(q)_{primal} = G - P_q$, respectively with the dual





functional $CC(h)_{dual} = J - P_h$.

$$\underset{\mathbf{q}}{\text{minimize}} \quad C(\mathbf{q})_{primal} = \sum_{i=1}^{n_{pipes}} \int_0^{q_i} \left[ rq|q|^{\alpha-1} - \left(A_f^T h_f\right)_i \right] dq \tag{7}$$

subject to $\quad \mathbf{Aq} + \mathbf{d} = \mathbf{0}$.

$$\underset{\mathbf{h}}{\text{minimize}} \quad CC(\mathbf{h})_{dual} = \sum_{i=1}^{n_{pipes}} \int_0^{\left(\mathbf{A}^T\mathbf{h} - \mathbf{A}_f^T\mathbf{h}_f\right)_i} r^{-\frac{1}{\alpha}} h|h|^{\frac{1-\alpha}{\alpha}} \, dh + \sum_{j=1}^{n_{nodes}} d_j h_j. \tag{8}$$

The major benefit of this optimization is that the existence of the solution is guaranteed due to the fact that the argument is convex and uniqueness if the argument is strictly convex as long as the set of feasible solutions is not empty. The prove for this has been given by Birkhoff (1963).

## 2.3 Solution Algorithms

Carpentier et al. (1985) published a comprehensive study on the efficiency of numerical solution strategies. The Hardy-Cross
algorithm is given by a relaxed Newton method that solves the equations successively for each loop. An approach to the parallel solution of the primal system is the Conjugate Gradient method, but due to the condition of the multidimensional optimization problem it may become inefficient for approaching the optimal point. In general Newton methods are more efficient in solving the equations as they benefit from quadratic convergence near the solution. Due to the better conditioning of the Hessian matrix the Primal-Newton method needs approximately half the number of iterations compared to the Dual-Newton method.
The most efficient solution algorithm is presented by hybrid algorithms that solve for the primal and dual variables like the one introduced by Hamam and Brameller (1971).

## 3 Pressure Driven Modeling

The demand driven modeling approach has considerable shortcomings under complex boundary conditions and is not able to realistically model mechanisms that are driven by pressure differences. Two of the most important phenomena are pressure
dependent demands at consumption nodes and pressure dependent leakage for pipe ruptures.

### 3.1 Pressure Driven Boundary Conditions

In the case of pressure dependent demand, experience has shown that under certain conditions the Demand Driven Model can lead to non-physical solutions. This is the case in pressure deficient networks where, under realistic conditions, the demand cannot be met at certain consumer nodes. From hydrostatics it is known that the maximum flow volume depends on the
difference between the nodal and the atmospheric pressure. To take this into account the Pressure Driven Modeling approach loosens up the demand boundary conditions and the fixed consumption is replaced by the set of inequality conditions $0 \le d \le$





$d_s$. They state that the actual discharge at the node lays in between zero and the desired service demand based on the state of the network.

By far the most popular approach to handle the degree of freedom introduced by the pressure dependent formulation is the introduction of a emitter function $c(h)$ that quantifies discharges based on the present head.

One of the first publications on the topic by Bhave (1981) uses the Heaviside function as emitter function. This means that for a head lower than the nodes elevation there is no supply and above the full demand is met. Wagner et al. (1988) introduced a more realistic model based on the hydrostatic equations for a free flow boundary condition. This leads to the Wagner function:

$$c(h) = \begin{cases} 0 & \text{, if } h \leq h_m \\ \left(\frac{h-h_m}{h_s-h_m}\right)^{\frac{1}{2}} d & \text{, if } h_m \leq h \leq h_s \\ d & \text{, if } h_s \leq h. \end{cases} \tag{9}$$

This equation complies with the inequality conditions and is based on a physical model for the outflow. In equation (9) $h$

is the calculated head. The minimum head necessary is given by $h_m$. In general the minimum head is defined by the nodal elevation. The required head for servicing the full requested demand is defined by $h_s$. Although the definition of an emitter function is a popular way of dealing with the challenge, there exist approaches for pressure dependent demand without the specific definition of emitter.

    In the case of pressure dependent discharges for pipe ruptures Schwaller and Van Zyl (2014) describe a concept called Fixed

and Varied Area Discharge (FAVAD) which defines an orifice function as follows.

$$c(h) = C_d \sqrt{2g} \left( A_0 h^{0.5} + m h^{1.5} \right) \tag{10}$$

Here $C_d$ is a discharge coefficient, $g$ the gravitational constant, $A_0$ is the area of the opening if no head is present and $m$ is a linear value describing the growth in leak area. In contrast to the demand emitter function, this orifice function not only defines the discharge based on the pressure but also the change in area for the rupture due to the elasticity of materials and the pressure.

## 3.2   Pressure Driven Problem Description

Using a general emitter function a modified set of equations has been published by Piller et al. (2003).

$$\mathbf{A}\mathbf{q} + \mathbf{c}(\mathbf{h}) = \mathbf{0} \tag{11}$$

$$\Delta\mathbf{h}(\mathbf{r},\mathbf{q}) - \mathbf{A}^T\mathbf{h} - \mathbf{A}_f^T\mathbf{h}_f = \mathbf{0} \tag{12}$$

    As for the Demand Driven Model it is also possible to formulate the Content and Co-Content problem for the Pressure

Driven approach. Following Piller et al. (2003) the Content model is defined by:

$$\underset{\mathbf{q}}{\text{minimize}} \quad C(\mathbf{q})_{primal} = \sum_{i=1}^{n_{pipes}} \int_0^{q_i} \left[ rq|q|^{\alpha-1} - \left(A_f^T h_f\right)_i \right] dq + \sum_{j=1}^{n_{nodes}} \int_0^{(-\mathbf{A}\mathbf{q})_i} c_j^{-1}(d)\, dd \tag{13}$$

$$\text{subject to} \quad \mathbf{0} \leq \mathbf{A}\mathbf{q} \leq \mathbf{d}.$$

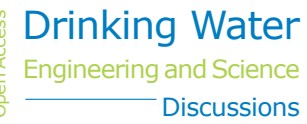

The pressure dependent Co-Content model is given as:

$$\underset{\mathbf{h}}{\text{minimize}} \quad CC(\mathbf{h})_{dual} = \sum_{i=1}^{n_{pipes}} \int_{0}^{\left(\mathbf{A}^T\mathbf{h} + \mathbf{A}_f^T\mathbf{h}_f\right)_i} r^{-\frac{1}{\alpha}} h \left|h\right|^{\frac{1-\alpha}{\alpha}} dh + \sum_{j=1}^{n_{nodes}} \int_{h_j^m}^{h_j} c_j(h) \, dh. \tag{14}$$

## 4 Deficient Networks

From literature the notion of deficient networks can take a number of different definitions. These definitions may be divided into model, mathematical and physical deficiencies. Model deficiencies are errors in the creation, conversion or transfer of the network graph. A mathematical deficiency can be defined as a maximal connected network where, due to some boundary condition the set of feasible solutions is reduced to the empty set or the solution is not unique. In contrast to mathematical deficiencies, in the case of a hydraulic deficiency a unique solution exists, but it is physically incorrect. With respect to the two modeling paradigms presented in chapters 2 and 3 different phenomena have to be classified as deficient. In the following a number of deficiency phenomena of special interest for the ResiWater project are presented and evaluated with respect to demand and pressure driven modeling.

**Conflicting constraints:** The first scenario is given if the boundary conditions are in conflict for certain parts of the network. This occurs if flow regulating devices are incorporated into the model and introduce additional constraints to the mathematical model. In unfortunate cases these constraints may conflict with the demand request of the consumption nodes. Simply put, the flow entering a region of the network is not satisfying the required demand. In demand driven modeling this reduces the set of feasible solutions for the optimization problem defined in equations (7) or (8) to zero as demonstrated by Deuerlein et al. (2012). Deuerlein also suggests an algorithm to determine if a feasible solution exists for the particular scenario. Looking at the pressure dependent calculation of the same system, it can be shown that by loosening the demand boundary conditions the system becomes solvable again, but the consumers will be supplied with a reduced flow.

**Ambiguous constraints:** Another example for a mathematical deficiency is given if the boundary condition allow for an infinite number of solutions. In their article Gorev et al. (2016) describe a scenario where two flow control valves (FCV) are installed in series. In this case the two FCVs create a combined head-loss, but due to the ambiguous nature of this problem an infinite number of solutions exist and it is impossible to determine which of the two FCVs contributes how much. This phenomenon is neither addressed by DDM nor by PDM approaches.

**Pipe rupture:** In respect to resilience, phenomena like pipe-rupture or -bursts are of special interest. In this cases the massive water loss dominates the flow in the network. Recent research has shown that the Fixed and Varied Area Discharge (FAVAD) model as described in equation (10) provides a good description for leakage behavior of elastic materials Van Zyl et al. (2011). Due to the pressure dependent nature of the phenomenon in demand driven modeling it is not possible to adequately handle the problem. In contrast, similar to the pressure driven demand, it is possible to solve these problems in the PDM framework.

**Low pressure zones:** The fourth scenario is correlated with the occurrence of low pressure zones in the network. This may for instance be triggered by a pipe burst and the subsequent pressure loss. Looking at current demand and pressure driven





models this behavior is not taken into account. In the case of zero or negative pressure, software packages like Porteau and EPANet will give a warning notifying the user that pressure dropped below zero, but the hydraulic connection is still intact and disconnected network parts will still be supplied. A conceptually simple way to solve this problem in the PDM framework may be implemented by an iterative approach that analyzes the pressure on every node and deletes all links connected to the

5    deficient ones. A different approach has been proposed by Piller and Van Zyl (2009). They introduce an additional constraint to the optimization formulation described in section 3 that reduces the flow on deficient pipes to zero.

## 5    Conclusions

In conclusion, this paper gave a summary on the current state of water distribution network modeling, looking into the classical approaches using mass balance and energy equations, as well as optimization approaches that allow to make assumptions for

10    the properties of the solution space. These formulations have been given for both the framework of demand driven modeling with very strict constraints and pressure driven modeling where the constraints have been relaxed to give more realistic results.

Looking at the two modeling paradigms, four phenomena of deficient networks of interest to the ResiWater Project have been analyzed with respect to solvability and physical accuracy. It could be concluded that, although the pressure driven approach is far superior to demand driven modeling in cases like pressure controlled demands and leakage, there still exist

15    model deficiencies in cases where pressure drops below a physically realistic level.

*Acknowledgements.* The work presented in the paper is part of the French-German collaborative research project ResiWater that is funded by the French National Research Agency (ANR; project: ANR-14-PICS-0003) and the German Federal Ministry of Education and Research (BMBF; project: BMBF-13N13690).





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
