# Peer review of "Limitations of Demand- and Pressure-Driven Modeling for Large Deficient Networks"

_Drinking Water Engineering and Science, 2017_

## Referee Comment (RC1) · Anonymous Referee #1 · 19 May 2017

The paper is clear and well written. I like the fact that it gives a framework of the state of the art of WDN modelling. However, I think that it misses some important reference such as: Todini E., Rossman LA "Unified Framework for Deriving Simultaneous Equation Algorithms for Water Distribution Networks." JOURNAL OF HYDRAULIC ENGINEERING-ASCE, Volume: 139, Issue: 5, Pages: 511-526.

Furthermore, how does the Authors' formulation relate to Todini and Pilati's GGA? And to other algorithms currently used in simulation software!

---

## Referee Comment (RC2) · Anonymous Referee #2 · 22 May 2017

The paper is aimed at providing a summary on the current state of water distribution network modeling. Unfortunately, the manuscript misses to take into considerations fundamental approaches developed for DD and PD simulation such as the global gradient algorithm and the several related papers recently published such as Giustolisi et al., 2008 and Todini and Rossman 2013.

Giustolisi, O., Savic, D. A., and Kapelan, Z. (2008). "Pressure-driven demand and leakage simulation for water distribution networks." J. Hydraul. Eng., 134(5), 626–635. Todini, E. and Rossman, L. 2013. Unified Framework for Deriving Simultaneous Equation Algorithms for Water Distribution Networks. J. Hydraul. Eng., 139(5), 511–526

---

## Referee Comment (RC3) · Anonymous Referee #3 · 31 May 2017

**Journal: DWES**
**Title: Limitations of Demand- and Pressure-Driven Modeling for Large Deficient Networks**
**Author(s): Mathias Braun et al.**
**MS No.: dwes-2017-13**
**MS Type: Research article**
**Special Issue: Computing and Control for the Water Industry, CCWI 2016**

The paper deals with two dual classical approaches regarding water distribution network modeling. They are dual approaches using balance equations (mass and energy), and optimization (energy). These approaches enable various assumptions for the properties of the solution space and are formulated with a stress on pressure driven modeling hypotheses, which use relaxed constraints thus producing more realistic results. The authors conclude that, although the pressure driven approach is far superior to demand driven modeling in a number of very well documented cases, various model deficiencies, for example, in cases where pressure drops below a physically realistic level, still persist.

The manuscript is on a very interesting topic, since management of deficient networks is currently paramount. As a result, the paper is well-suited for the Journal. The organization of the manuscript is in a good shape. The methodology is well documented and the results are convincing. In my view, however, the literature review in the introduction should be improved. Are there no advances in the subject produced during the last decade?

Additionally, I recommend that the authors revise the paper for some minor errata. Also, English writing, though perfectly understandable, should be carefully revised for better readability.

Examples:

Line 9: '…represents pipe sections as a link and…' should be 'represents pipe sections as links and…'

Check the use of commas and/or dots after equations. For example, around eqs (1) and (2) some commas are missing; the final dot in line 26 should be a comma,…

Line 9: replace 'chapter' for 'section'.

Line 26: FAVAD definition has already been given above.

After it, in my opinion the paper can be published without additional review. I believe the paper makes a valuable contribution through the analysis of existing methods and proposing interesting approaches.

---

## Short Comment (SC1) · 20 Jun 2017

June 2017

Journal of Drinking Water Engineering and Science

**Limitations of Demand- and Pressure-Driven Modeling for Large
Deficient Networks**

By
Mathias Braun, Olivier Piller, Jochen Deuerlein and Iraj Mortazavi

This paper explores the calculation of the state variables of flows and heads for water distribution systems. Both Demand Driven Modeling and Pressure Driven Modeling are considered. Equations are formulated using a content/co-content approach based on optimization to ensure existence and uniqueness of solutions.

The article provides an interesting review of DDM and PDM modeling. The most relevant and new material presented in the paper are the 4 issues raised in Section 4 of the paper. The authors highlight what the problems are but offer no direct solutions. This is clearly the subject of further research and hopefully will be presented in future papers.

**Minor Comments**

|  |  |
|---|---|
| I. | Page 1, Line 4 – replace 'cannot" with "may be" |
| II. | Page 1, line 5 – replace "look on" with "look at" |
| III. | Page 1, Line 21- last word – replace "done" with carried out" |
| IV. | Page 1, Line 23 – replace the word "tension" – this does not make sense in English in the context presented here |
| V. | Page 2, Line 4 – last word – replace "voiced" with presented" |
| VI. | Page 2, Line 23 – replace "current on" with "current for" |
| VII. | Page 3, Line 26 – replace "to calculate" with "the calculation of" |
| VIII. | Page 4, Line 6 – replace "prove" with "proof" |
| IX. | Page 4, Line 14 – after first word "matrix" place a comma |
| X. | Page 4, Last Line of Page – replace "loosens up" by "relaxes" |
| XI. | Page 5, First Line of Page – replace "lays in" by "is in the range" |
| XII. | Page 5, Liner 5 – replace "as emitter" by "as an emitter" |
| XIII. | Page 5, Line 6 – replace "and above the" by "and for a head above a particular level" |
| XIV. | Page 6, Line 4 – replace "from literature" with "from the literature" |
| XV. | Page 6, Line 9 – replace "chapters" with 'sections" |
| XVI. | Page 6, Line 18 – replace 'loosening" with 'relaxing' |
| XVII. | Page 6, Line 20 – 3rd last word – replace "allow" with allows" |
| XVIII. | Page 6, Line 25 – Replace "In respect' with 'With respect" |
| XIX. | Page 7, Lines 1 and 2 – give references for the two software packages that are mentioned |
| XX. | Page 7, Line 8 – replace 'this paper gave" with "this paper has given" |
| XXI. | Page 7, Line 13 – replace "It could" with "It can" |
| XXII. | Page 8 – move top 2 references down to be alphabetical |
| XXIII. | Page 8, Bhave reference – remove journal of ASCE |
| XXIV. | Page 8, Line 15 – give complete reference |
| XXV. | Page 8, Line 16 – the Elhay reference is not mentioned in the paper at all – it should be included |

---

## Author Comment (AC1) · 22 Jun 2017

Dear Angus,

thank you very much for spending the time to review the article. Your comments are very justified and helpfull. I will change the article accordingly.

Kind regards, Mathias

P.S.: Thanks for polishing up my english.

---

## Author Comment (AC2) · 30 Jun 2017

Dear Reviewer #1,

thank you very much for spending the time to review the article and the kind evaluation. The article by Todini and Rossmann gives a recent review and classification on solution algorithms for the Demand Driven Model that complements the paper by Carpentier very well. It has been added to the literature in chapter on DDM. Also in the chapter on DDM we added a sentence on the GGA, which can be interpreted as a Newton method applied to equation 5 in the article.

Best regards, Mathias Braun

---

## Author Comment (AC3) · 30 Jun 2017

Dear Reviewer #2,

We thank you very much the time you spent on the review of the article.

The background on pressure driven demand and leakage formulation was extended by the article by Giustolisi et al. 2008. In addition, a reference to the paper by Todini and Rossmann was added.

However, we respectfully disagree with the assessment of the articles aim. The aim of the article was not to give a summary on the current state of water distribution network modelling. This has been done by others in much more detailed contributions, the publications mentioned by the reviewer included. Instead the aim of this article is to

explore the abilities and limitations of those modeling approaches concerning a number of demanding scenarios.

Kind regards, Mathias

––––––––––––––––––––––––––––––

---

## Author Comment (AC4) · 30 Jun 2017

Dear Reviewer #3,

thank you very much for the kind review of the article and the time you invested into it.

For the literature review to our knowledge there have been no advances in the variational /optimization formulation since the contributions of Birkhoff, Collins and Carpentier. For the Pressure Driven Model we will extend the literature by references to the recent paper by Giustolisi et al. on pressure driven demand and leakage simulation. Also we will add additional literature on advances in loop modeling and partitioning algorithms to chapter 2.

Further, the minor changes you proposed have been addressed in the revised

manuscript.

Kind regards, Mathias Braun
* * *